

# The PRISM4 (mid-Piacenzian) palaeoenvironmental reconstruction

Harry Dowsett[1], Aisling Dolan[2], David Rowley[3], Matthew Pound[4], Ulrich Salzmann[4], Marci Robinson[1], Mark Chandler[5,6], Kevin Foley[1], Alan Haywood[2]

[1]Eastern Geology and Paleoclimate Science Center, United States Geological Survey, Reston VA 20192 USA
[2]School of Earth and Environment, University of Leeds, Leeds, LS2 9JT UK
[3]Department of the Geophysical Sciences, University of Chicago, Chicago, IL, 60637 USA
[4]Department of Geography, Faculty of Engineering and Environment, Northumbria University, Newcastle upon Tyne NE1 8ST, UK
[5]Center for Climate Systems Research at Columbia University, New York, NY USA
[6]NASA Goddard Institute for Space Studies, New York, NY USA

*Correspondence to*: Harry Dowsett (hdowsett@usgs.gov)

**Abstract.** The mid-Piacenzian is known as a period of relative warmth when compared to the present day. A comprehensive understanding of conditions during the Piacenzian serves as both a conceptual model and a source for boundary conditions and means of verification of global climate model experiments. In this paper we present the PRISM4 reconstruction, a palaeoenvironmental reconstruction of the mid-Piacenzian (~3 Ma) containing data for palaeogeography, land and sea-ice, sea-surface temperature, vegetation, soils and lakes. Our retrodicted palaeogeography takes into account glacial isostatic adjustments and changes in dynamic topography. Soils and lakes, both significant as land surface features, are introduced to the PRISM reconstruction for the first time. Sea-surface temperature and vegetation reconstructions are unchanged but now have confidence assessments. The PRISM4 reconstruction is being used as boundary condition data for the Pliocene Model Intercomparison Project, Phase 2 (PlioMIP2) experiments.

## 1 Introduction

The Pliocene, specifically the mid-Piacenzian (3.264 Ma to 3.025 Ma), has been a focus of synoptic palaeoclimate research for the past 25 years. The mid-Piacenzian is the most recent time in Earth's past to have exhibited climates not unlike those projected for the end of the 21st century (Dowsett, 2007a; IPCC, 2013). With widespread recognition by most experts that anthropogenic drivers have been the dominant cause of observed warming since the mid-20th century (Verheggen et al., 2014), and surface temperatures projected to rise during the 21st century under all emission scenarios (IPCC, 2013), understanding the Pliocene climate has taken on new importance. While not a temporal analog to future climate conditions, there is much to learn about the magnitude and spatial distribution of processes from this, in essence, natural climate laboratory (Crowley, 1990).

Borrowing heavily from methodology used by the CLIMAP (Climate / Long-Range Investigation, Mapping and Predictions) for the reconstruction of the last glacial maximum (LGM), the U.S. Geological Survey (USGS) and collaborators initiated a



large-scale data collection and interpretation project: Pliocene Research, Interpretation and Synoptic Mapping (PRISM) (see Dowsett, 2007a; CLIMAP, 1976). The first PRISM reconstructions consisted of data sets for sea-surface temperature (SST), vegetation, land ice distribution and volume, sea ice cover, land elevation and sea level (Dowsett *et al.* 1994; 1999). Since its inception in the late 1980s, PRISM has grown in both size and scope, evolving through 4 global scale reconstructions, each

one adding a new component of the Piacenzian palaeoenvironment or improving upon methods to increase confidence in the reconstructions (see Dowsett, 2007; Dowsett *et al.*, 1994; 1996; 1999; 2010; 2013). These reconstructions serve two purposes: to assemble the best information possible to provide a conceptual model of the Piacenzian palaeoenvironment *and* to provide the data as quantitative, gridded arrays to the palaeoclimate modeling community for global climate model simulations.

Early modeling efforts to simulate a warmer Pliocene Earth used atmospheric general circulation models (AGCMs) initiated

with higher than pre-industrial values of $CO_2$ in the atmosphere (Chandler et al., 1994; Dowsett, 2007b; Haywood et al., 2000; Sloan et al., 1996). These simulations used prescribed boundary conditions in the form of PRISM synoptic reconstructions of SST, land cover, and topography (Dowsett et al., 1996; Dowsett et al., 1999; Dowsett, 2007a).

PRISM3 was the basis for the Pliocene Model Intercomparison Project (PlioMIP), the first Pliocene multi-model comparison using fully coupled atmosphere-ocean general circulation models (AOGCMs) (Haywood et al., 2010; 2011). Much effort was

put into improving the land surface cover and ocean components for this penultimate reconstruction (Dowsett *et al.*, 2010). PRISM3 introduced a multiproxy SST reconstruction, a deep ocean temperature reconstruction, and for the first time a biome classification of the Pliocene land surface.

The PRISM3 reconstruction also addressed uncertainty in both terrestrial and marine palaeoclimate estimates (Dowsett et al., 2012, 2013a; Salzmann et al., 2013). Marine and terrestrial data/model comparisons (DMC) presented in IPCC AR5

documented robust large scale features of the Piacenzian climate but at the same time identified areas of discord between data and models, highlighting the need for additional assessments of confidence for both reconstructed and simulated environments (Haywood et al., 2013; Salzmann et al., 2013; Dowsett et al., 2012; 2013a).

PRISM data have also been used to study diversity and ecological niche changes in planktic foraminifers, mollusks and fish in the face of profound global warming (e.g., Yasuhara et al., 2012; Saupe et al., 2014, Jacobs, 2015).

In this paper we document and summarize our most recent reconstruction, PRISM4. PRISM4 is a conceptual model of mid-Piacenzian conditions for which major efforts have been focused on improving the palaeogeographic and cryospheric components. New topography/bathymetry, Greenland ice sheet (GIS), soil, and lake distributions are discussed, including those PRISM4 features being used as boundary conditions for climate model simulations. Confidence in palaeoenvironmental reconstruction is receiving much interest in the palaeoclimate modeling community, and changes necessary to address

uncertainty are discussed. We also discuss both terrestrial and marine high-resolution data being developed as part of PRISM4 to better understand Piacenzian palaeoclimate variability.





## 2 PRISM Chronology

Previous PRISM palaeoclimate reconstructions were based upon a *time slab* concept developed early in the project to overcome the then inability to make long distance high-resolution correlations (Dowsett and Poore, 1991). This was achieved for SST data and land surface cover reconstruction, by averaging the warm phase of climate or selecting the vegetation representing

the wettest and warmest period within the defined temporal slab at each locality (Dowsett et al., 2009; 2013; Salzmann et al., 2008; 2013). The PRISM reconstruction therefore approximates the average "interglacial" condition at each site. The initial slab was a ~300 Ky interval ranging from within the Mammoth to just above the Kaena reversed Polarity Subchrons of the Gauss normal Polarity Chron (Figure 1). This and a number of biochronologic events allowed approximate identification of the slab position in both marine and terrestrial settings (see Dowsett, 2007a and references cited therein).

The PRISM3 *time slab* (Dowsett *et al.*, 2010) was refined to the interval between 3.264 Ma and 3.025 Ma, easily identified in marine sequences by its position between oxygen isotope enrichments M2 and G20 (Figure 1) in the LR04 chronology of Lisiecki and Raymo (2005). The 3.264 Ma to 3.025 Ma range is still appropriate for most components of the PRISM4 reconstruction (see discussion).

New PRISM4 time-series data use a nested chronology since various components of the palaeoenvironmental reconstruction

can achieve different maximum resolution. PRISM4 marine time series are generated from localities possessing the characteristics needed for precise orbital scale correlation. The challenges of terrestrial age dating and stratigraphic control of many terrestrial archives currently limit expansion of the terrestrial reconstruction of PRISM4 time series in all but a few locations.

The PRISM time slab or PRISM "interval" as defined above, occurs within the Piacenzian Age.  The Piacenzian is roughly

equivalent to the Gauss normal polarity Chron (~3.6 Ma to 2.6 Ma). Prior to 2010 the Pliocene Epoch included the Zanclean, Piacenzian and Gelasian Ages. Thus it was common practice to refer to the PRISM interval as the mid-Pliocene. Changes enacted by the International Commission on Stratigraphy revised the placement of the Pliocene – Pleistocene boundary from the base of the Calabrian Stage (1.801 Ma) to the base of the Gelasian Stage (2.588 Ma) (Gibbard et al., 2010). This change makes it awkward to refer to the mid-Piacenzian PRISM interval as mid-Pliocene (see also Dowsett and Caballero-Gill, 2010).

Previous publications referring to the PRISM interval, PRISM time slab, mid-Pliocene warm period (mPWP) or mid-Piacenzian all refer to the same interval of time originally defined by Dowsett an Poore (1991) and revised by Dowsett et al. (2010) as discussed above (Figure 1).  We propose the term mid-Piacenzian be used hereafter.

## 3 PRISM4 Piacenzian Palaeoclimate Reconstruction

PRISM4 contains internally consistent and integrated data sets representing our best synoptic understanding of

palaeogeography, sea level, ocean temperature, terrestrial vegetation, soils, lakes and land and sea-ice distribution (Table 1). Some PRISM4 data sets received only minor alterations from PRISM3 while others represent fundamental changes in methodology or add additional facets of the palaeoenvironment not explicitly included in earlier versions of our reconstruction.



The single largest change accompanying the PRISM4 reconstruction is the addition of our palaeogeography (topographic and bathymetric data) (see Rowley et al., 2013). These data are integrated with Greenland and Antarctic ice sheets (see Section 3.1), which are in agreement with the estimates of mid-Piacenzian sea level (Miller et al., 2012). Our land cover analysis has been enhanced with new data sets depicting the spatial distribution of lakes and soil types (Pound et al., 2014). SST and

terrestrial vegetation cover are essentially unchanged from PRISM3, but confidence schemes have been created to assess individual sites used for the reconstruction (Salzmann et al., 2013, Dowsett et al., 2013). In addition, for the first time, new PRISM4 high-resolution time series are being developed to *specifically* address variability within the PRISM time interval. The interrelationships and dependencies between the PRISM4 and PRISM3 reconstructions are illustrated in Figure 2. Major components of the reconstruction are briefly discussed in the following sections, and data are available here:

http://geology.er.usgs.gov/egpsc/prism/4_data.html

Readers are referred to the original citation of each data set for more detailed background and descriptions of methodologies (Table 1).

### 3.1 Ice Sheets

Direct evidence for Piacenzian ice sheets (Greenland and Antarctica) is sparse. Therefore, the size, placement and volume of

ice in polar regions must be remotely assessed by a number of other techniques (e.g., proximal marine records of ice rafted detritus, marine oxygen isotope records combined with bottom water temperature estimates and indicators of sea ice). These techniques constrain global and possibly regional ice volume but not its three-dimensional expression. Ice sheet models can be used to generate potentially realistic configurations based upon all available evidence.

The PRISM4 GIS is based upon results from the Pliocene Ice Sheet Modeling Project (PLISMIP) presented in Koenig et al.

(2015) and Dolan et al. (2015a). This GIS configuration (Figure 3) represents our highest-confidence for the possibility of ice sheet location during the warmest parts of the mid-Piacenzian and is constrained by available proxy data. The PRISM3 GIS was centrally located and covered approximately 50% of the area of the present day ice sheet (Dowsett et al., 2010). In contrast, the PRISM4 GIS is confined to high elevations in the Eastern Greenland Mountains covering an area approximately 25% of the present day GIS.

The dynamic nature of the Antarctic cryosphere during the Pliocene has been debated in the literature for decades and, as with the GIS, direct evidence for ice conditions is limited to the Dry Valleys and proximal marine records (e.g., Shackleton and Kennet, 1975; Webb et al., 1984; Dowsett and Cronin, 1990; Prentice et al., 1993; Shackleton et al., 1995; Kennett and Hodell, 1995; Naish et al., 2009; McKay et al., 2012). McKay et al. (2012) document a dynamic ice sheet prior to 3.3 Ma followed by expansion of the ice sheet and concomitant SST cooling. Recent investigations of the Wilkes and Aurora subglacial basins

(East Antarctica) indicate both may have experienced a dynamic Pliocene ice history (Young et al., 2011).

The PRISM3 Antarctic ice reconstruction was generated using the British Antarctic Survey Ice Sheet Model (BASISM), driven by an AGCM experiment using prescribed PRISM2 boundary conditions (Hill et al. 2007; Hill, 2009; Dowsett et al., 2010). In the PRISM2 reconstruction, the Wilkes and Aurora subglacial basins had reduced areal extent of ice sheets relative to the





present day. PRISM3 however removed the WAIS based upon available information that suggested there was no ice present during the warmest parts of the Pliocene (Naish et al., 2009; Pollard and DeConto, 2009; Hill et al., 2010). Since the EAIS configuration used in PRISM3 is still supported by recent work, we have retained the PRISM3 Antarctic ice configuration for the PRISM4 reconstruction (Figure 3).

## 3.2 Palaeogeography

Modeling the dynamic nature of the Earth's surface over time is an area of increasing interest and complexity. It has become apparent that a physically-based methodology for deriving Piacenzian global topography is necessary and that it needs to account for first order effects of mantle convection (dynamic topography) and to a lesser extent, glacial isostatic adjustment (GIA) (Rowley et al., 2013).

PRISM3/GISS topography was based upon the palaeogeographic maps of Markwick (2007) (Sohl et al., 2009). For PRISM4 we have developed a new procedure to retrodict the palaeogeography of the Piacenzian taking into account major processes that affect surface topography and sea level as a function of time. The main processes considered are (i) changes in dynamic topography associated with mantle flow and (ii) glacial isostatic adjustment (GIA) due to Piacenzian ice loading. We also calculate a global land-sea distribution (sea level) prediction based on the reconstructed Piacenzian topography. Essentially the retrodicted PRISM4 Piacenzian palaeogeography is the result of a number of steps detailed below (and summarized in Figure 4). The palaeogeography is constrained to agree with global integral properties of the Earth, such as the conservation of the Earth's mean radius and the preservation of the volume of water at the surface of the Earth.

### 3.2.1 Initial base topography

Our starting conditions are based on the ETOPO1 (1 arc-minute) global relief model of the Earth's surface that integrates land topography and ocean bathymetry (Amante and Eakins, 2009). Figure 4a shows the 'bedrock' (base of the ice sheets) variation of ETOPO1 sampled at the working resolution of a quarter degree. As no glacial isostatic adjustment (GIA) correction has been applied to this version of ETOPO1, initially we apply a GIA correction to account for both the remaining disequilibrium due to the ongoing relaxation of topography from Last Glacial Maximum (LGM) ice loads (Raymo et al., 2011; Figure 4b) as well as all Present day ice and GIA associated with melting of this ice. This results in a nominal equilibrium topography that would pertain in ~200 kyr, assuming all ice and ice-related loading have been accounted for. There are however features that are prominent 'fingerprints' of Pleistocene glaciations, such as the Great Lakes and the Canadian Archipelago, that were not features of the landscape during the Piacenzian (Figure 4c). Therefore, the rebounded topography reconstruction shown in Figure 4b was contoured at 50m intervals and then the Pleistocene landscape features were filled in to achieve an approximately uniform average elevation in each region (Figure 4d). In reality the sediments eroded from areas such as the Great Lakes, will have been redeposited elsewhere, for example in the Gulf of Mexico (Galloway, 2008). A more consistent method of applying a Pleistocene fill would be to take sediment from known depositional environments and redistribute it in the erosional settings.



However, we have chosen to implement a more simplistic representation as there are currently no detailed isopachs of preserved sediment thickness or reconstructions of denudation to insure that our model agrees with observations.

### 3.2.2 Dynamic topography

Dynamic topography is defined as the response of the Earth's surface to the radial component of stresses originating from
buoyancy residing in the mantle and the lithosphere (Forte et al., 1993). It is the component of topography that is not explained by variations in crustal thickness and/or density and can be considered as the difference between present-day topography and the crustal isostatic topography.

In order to compute an estimate of dynamic topography change since the Piacenzian we use a calculated global mantle flow field (Moucha et al., 2009; Rowley et al., 2013). This calculation is based on the use of joint seismic tomography and
geodynamic inversions that map the spatial distribution of density in the mantle through a simultaneous inversion of seismic travel-time data with various geophysical observables (Forte et al., 2010; Simmons et al., 2006; Simmons et al., 2009). These features include global air-free gravity anomalies, crust-corrected estimates of dynamic surface topography, plate divergence and the dynamic ellipticity of the core-mantle boundary (Forte et al., 2010; Simmons et al., 2006; Simmons et al., 2009). The mantle flow calculations (e.g. Forte, 2007; Forte, 2011; Forte et al., 1993; Forte et al., 2010; Moucha et al., 2008a, 2008b ;
Moucha et al., 2009) are formulated for a fully compressible, self-gravitating mantle with a mean density profile given by PREM (Dziewonski and Anderson, 1981) and geodynamic inferences of the depth variation of the mantle viscosity based on Mitrovica and Forte (2004).

For the dynamic topography correction shown in Figure 4e, we use the mantle viscosity model V2 (Moucha et al., 2008) and the seismic tomography model TX2008 (Forte et al., 2010) that represent a sensible realization of the dynamic topography
change contributions to Piacenzian topography. Our choice of seismic tomography-derived density distribution and the mantle viscosity model will affect both the amplitude and location of our dynamic topography estimate and the rate of its change over time (Rowley et al., 2013), but not significantly on the scale of the quarter degree resolution of the reconstruction. Application of the dynamic topography correction results in the palaeogeography shown in Figure 4f.

### 3.2.3 Glacial Isostatic Adjustment

Glacial Isostatic Adjustment (GIA) refers to the deformational, gravitational and rotational adjustment of the Earth as a consequence of changes in ice sheet loading (Farrell and Clark, 1976; Mitrovica and Milne, 2002). The present topography of the Earth is still significantly impacted by the loading and unloading of ice during the Late Pleistocene glacial cycles. The current grounded ice sheets of Greenland and Antarctica also affect the global surface topography. Finally, the potential ice sheet loading based on our defined mid-Piacenzian ice sheets (Figure 4g, see above) will also have an impact on the
reconstructed topography. Following methods employed in Raymo et al. (2011) and Rowley et al. (2013) and based on the seminal paper of Farrell and Clark (1976), we incorporate a full GIA correction into our PRISM4 palaeogeography (Figure 4h) thus taking into account the largest factors that influence topography over time.



Figure 5 shows the final land-sea distribution of the PRISM4 palaeogeography. The major differences from the PRISM3 palaeogeography of Sohl et al. (2009), PRISM2 topography of Dowsett et al. (1999), and the present day geography, are a closed Bering Strait and closed Canadian Arctic Archipelago, effectively closing off the Atlantic connection to the Arctic through the Labrador Sea and Baffin Bay. The PRISM4 mid-Piacenzian Indonesian region is elevated compared to present. Shoaling in this region would have created a more restricted seaway between the Pacific and Indian Oceans with potential effects on circulation and heat exchange with the atmosphere (Godfrey, 1996). See discussion for a complete analysis of the palaeogeographic differences between PRISM4 and previous PRISM topographies.

### 3.3 Sea level

Piacenzian sea-level estimates in the literature range from no change to almost +50m above present day based upon a variety of techniques (e.g., Dowsett and Cronin, 1990; Wardlaw and Quinn, 1991; Brigham Grette and Carter, 1992; Kennett and Hodell, 1995; Miller et al., 2005, 2012; Sosdian and Rosenthal, 2009; Naish and Wilson, 2009; Dwyer and Chandler, 2009; Winnick and Caves, 2015). Early PRISM sea level estimates were primarily glacioeustatic and dealt with tectonic subsidence and uplift in a relatively simplistic fashion (Dowsett and Cronin, 1990). Sea level changes must also take into account a dynamic topography, the result of glacial isostatic adjustments (GIA) combined with changes due to mantle convection (Rowley et al., 2013). PRISM3 qualitatively compared ranges of sea level estimates from a number of sources and found a sea level rise of +25m relative to present day fit all available data (Dowsett et al., 2010). In a more recent and quantitative treatment of available data, Miller et al. (2012) compared sea level changes derived from backstripping studies in Virginia, New Zealand and Enewetak Atoll with those from benthic $\delta^{18}O$ data and Mg/Ca estimates. They discussed the limitations of various methods and used the data to derive an empirical estimate of uncertainty for individual sea level estimates. They concluded that a sea level of +22m ±5m was *likely* (68% confidence interval) and +22m ±10m *extremely likely* (95% confidence interval), relative to present day sea level, to have been reached within the PRISM interval.

The Miller et al. (2012) estimate implies substantial reduction in mid-Piacenzian global ice volume, equivalent to the combined volumes of the present day Greenland and West Antarctic Ice Sheets (GIS and WAIS, respectively). They postulated it likely that several meters could be attributed to the margins of the East Antarctic Ice Sheet (EAIS). The reconstructed PRISM4 ice sheets have a total volume of 20.1 x $10^6$ km$^3$. Relative to the modern observed volume of the Greenland (Bamber et al., 2013) and Antarctic ice sheets (Fretwell et al., 2013), we use the method described in de Boer et al. (2015) to estimate a sea level equivalent change for the mid-Piacenzian. Our ice sheet configurations equate to a sea level increase of less than ~24 m, without taking into account changes to the size of the global ocean in the Pliocene (as shown by changes in the land-sea mask shown in Figures 3 and 5). The PRISM4 sea level estimate differs slightly higher than the PRISM3 sea level estimate (based upon the PRISM3 ice sheets) due to the reduction in the size of the Greenland ice sheet relative to PRISM3 (Dowsett et al., 2010), but remains consistent with other available sea level data.



### 3.4 Ocean temperature and sea ice

Global fields of SST and Sea Ice (Figure 3) were described in Dowsett et al. (2009, 2010) and remain unchanged from the PRISM3 reconstruction. These fields are based upon multiple proxy analyses derived using a warm peak averaging (WPA) technique and time slab concept. While useful for a general synoptic view of mid-Piacenzian interglacial ocean conditions, the

increasing use of the PRISM reconstruction for data/model comparison requires a more refined temporal window. Decrease in the need to drive atmospheric GCMs with monthly temperature fields has led to only small enhancements to this portion of the PRISM data set (Figure 2). Instead, emphasis has been placed on new locality based, confidence-assessed, high-resolution time series (Dowsett et al., 2012; 2013a; Haywood et al. 2016; see Discussion section).

### 3.5 Terrestrial biomes

The PRISM4 Biome reconstruction (Figures 3 and 6a) has been updated from Salzmann et al. (2008, 2013) and Dowsett et al. (2010). Biomes are based upon data from 208 sites compiled from the literature (Salzmann et al., 2008). Surface temperature and precipitation anomalies have been derived from literature using the Coexistence Approach (Mosbrugger and Utescher, 1997) and multi-proxy temperature reconstructions. The relative confidence one can place on estimates from a particular locality was determined by Salzmann et al. (2013). In order to address climate variability, our biome map provides, for selected

palaeobotanical localities with sufficient data resolution, a biome reconstruction for a colder/drier and warmer/wetter interval within the Piacenzian stage (Figure 6a). Future PRISM terrestrial analysis, like the marine work, will shift focus to localities that allow for high-resolution chronology required to address magnitude and variability of climate within a much (temporally) reduced time slab.

### 3.6 Soils and Lakes

PRISM4 soil and lake distributions have been generated from a compilation of published literature (Pound et al., 2014). Nine soil types were identified from 54 palaeosol occurrences and recorded. Pound et al. (2014) related soils at these 54 localities (Figure 6b) to the PRISM3 biome reconstruction of Salzmann et al. (2008) and derived soil types for all land areas (Figure 3). Based upon color and texture each soil can be assigned an albedo value (Pound et al., 2014; Haywood et al., 2016). Lake data have been compiled from sedimentological evidence, dynamic elevation models, topographic studies, fauna, flora, or a

combination of these (e.g., Adam et al., 1990; Muller et al., 2001; Drake et al., 2008; Otero et al., 2009). Piacenzian lake surface area was taken from published estimates or from reconstructed lake extents. These surface areas were then translated into a percentage of a PRISM4 grid cell (2° latitude by 2° longitude), and where megalakes occupied more than one grid cell, the geographic distribution (See Figure 3 and 6b) was based upon the published distal-most latitude-longitude points and reconstructed shape (for full methodology see Pound et al., 2014).



## 4. Discussion

### 4.1 Areas of palaeogeographic change

#### 4.1.1 Marine

The Isthmus of Panama is above sea level in the PRISM4 reconstruction, effectively eliminating ocean circulation between
the Atlantic and Pacific via the Central American Seaway (CAS). While there is evidence for occasional brief post mid-Piacenzian breaches of the Isthmus based upon Caribbean salinity records (e.g., Coates et al., 1992; Cronin and Dowsett, 1996), the overall restriction of Caribbean and Pacific waters by the Isthmus was fully established by ~4.2 Ma (Haug et al., 2001; Montes et al., 2015). This CAS closure set up the present-day salinity contrast between the Atlantic and Pacific, shunting warm salty water northward along the east coast of North America as the Gulf Stream Current. The Gulf Stream diverges from
North America and becomes the North Atlantic Current, directing warm salty water toward the northeastern North Atlantic. De Schepper et al. (2009) related Marine Isotope Stage (MIS) M2 to a brief reopening of the CAS that weakened the North Atlantic Current and reduced northward ocean heat transport. Larger (compared to present day) ice sheets during MIS M2 (~3.3 Ma) are supported by a coupled ocean-atmosphere climate simulation targeted at this time interval (Dolan et al., 2015b), however this study did not test the hypothesis of an open CAS as a mechanism for ice sheet initiation. New PRISM4 surface
air and SST time series covering the MIS M2 through MIS KM5 interval should provide further information on marine and terrestrial conditions during and post MIS M2.

The Bering Strait initially opened in the late Miocene, possibly earlier, based upon the presence of the Atlantic bivalve *Astarte* occurring in sediments of the Alaskan Peninsula correlated to subzone b of the *Neodenticula kamtchatica* Diatom Zone (5.5 – 4.8 Ma) (Marincovich and Gladenkov, 1999; 2001).  Faunal exchange of thermophillic ostracodes between the Atlantic and
Pacific via the Arctic also supports an open Bering Strait during at least parts of the Pliocene and early Pleistocene (Cronin et al. 1993; Matthiessen et al., 2009). This was the rationale for an open seaway in the PRISM3 reconstruction. Our independent PRISM4 palaeogeographic analysis (described above) indicates a *closed* Bering Strait. The open or closed state of the Bering Strait, since the mid-Piacenzian, is dictated in large part by global eustatic and regional tectonic processes. Our decision to follow the palaeogeographic model is based upon the shallow depth of the seaway and evidence for repeated episodes of
subaerial exposure in both the Early Pliocene and during the Pleistocene (Hopkins, 1959; 1967; Nelson et al., 1974).

The effect of a closed Bering Strait has been shown to increase Atlantic Meridional Overturning Circulation (AMOC), which provides for an increase in ocean heat transport to the Arctic (Hu et al., 2010; 2015). Climate model simulations to date have been unable to achieve the high latitude warming estimated for both terrestrial and marine environments of the mid-Piacenzian (Salzmann et al., 2013; Dowsett et al., 2012; 2013a).  Hu et al. (2015) show that regardless of climate state, a closed Bering
Strait produces a warmer North Atlantic. Thus, a closed Bering Strait may improve the fit between existing Piacenzian marine and terrestrial data and model simulations.

Matthiessen et al. (2009) suggest the possibility of a closed Canadian Arctic Archipelago with Arctic – North Atlantic communication restricted to the Fram Strait. Evidence for this configuration was presented by Harrison et al. (1999) who





postulated establishment of an Early Pleistocene drainage system on underlying Pliocene deposits that drape the Canadian Arctic (Harrison et al., 1999) and the presence of fluvio-deltaic sedimentation around the margins of the Arctic continental shelf (Yorath et al., 1975; Jones et al., 1992). This fits well with the PRISM4 palaeogeographic model that shows most of the Canadian Arctic Archipelago to be subaerially exposed (Figures 3, 5).

The depths of the present day Gulf of Carpenteria, Arafura Sea, Java Sea, as well as sills in the region, are shallow (Smith and Sandwell, 1997). Our palaeogeographic model suggests regional tectonics produced even shallower conditions during the mid-Piacenzian, relative to present-day, from Southeast Asia to the Gulf of Carpenteria (Figures 3 and 7). A shallower than present Indonesian Throughflow (ITF) would alter regional sea-surface topography and restrict circulation, thus reducing heat transport from the Pacific to the Indian Ocean. Model simulations have shown that the result is an increase in SST and

precipitation in the western Pacific with an accompanying decrease in both SST and rainfall in the eastern Indian Ocean. Such changes in a dynamically sensitive region of the tropics can lead to altered atmospheric pressure and wind stresses (Schneider, 1998; Lee et al., 2002; Sprintall et al., 2015) with complex teleconnections capable of distributing the effects globally (Shukla et al., 2011).

### 4.1.2 Terrestrial

In addition to changes to important ocean gateways mentioned above, general differences exist between the new PRISM4 mid-Piacenzian and present-day palaeogeographies. Our reconstruction shows most continental areas at slightly higher elevations during the Piacenzian than they are today (Figure 7). In our mid-Piacenzian reconstruction, decreased ice loading of the continents plus increased water loading of the oceans, relative to present day, results in mantle displacement from beneath the oceans to beneath the continents, raising most continental elevations. Exceptions are areas immediately adjacent to the rift

valleys in Africa, the Sierra Madres of Mexico, the Andes north of Lima and south of Santiago, the Chugach in northwestern North America, the Tibetan Plateau, Sumatra and the northern part of the Great Dividing Range in Eastern Australia.

Most of Greenland was at a lower elevation during the mid-Piacenzian, relative to present day, due to the PRISM4 GIS being restricted to eastern Greenland (Koenig et al. 2015; Dolan et al. 2015a). Portions of Antarctica also show large elevation changes due to cryospheric changes implemented in the PRISM4 palaeogeography. These changes would have affected

atmospheric circulation, precipitation patterns and distribution of terrestrial vegetation.

### 4.2 Uncertainties in the ice sheet reconstruction

Both the Greenland and Antarctic ice sheets are reduced in terms of their volume and extent in the PRISM4 reconstruction relative to present day. As described previously there is little geological evidence that directly corroborates the exact extent and volume of the major ice sheets during the mid-Piacenzian. Therefore, the ice sheet configurations used in the PRISM4

reconstructions are plausible representations of potential mid-Piacenzian ice sheets. Nevertheless, they have inherent uncertainties associated with their reconstruction which should be noted. Recent studies by de Boer et al. (2015), Dolan et al. (2015a) and Koenig et al. (2015) have demonstrated unequivocally that ice sheet model reconstructions for Antarctica and



Greenland are both dependent upon the climate forcing employed and the choice of ice sheet model. As both of the reconstructions are based on modeled output, there will be some inherent model dependency in the PRISM4 reconstruction (e.g. if we had used a different model/climate forcing, the configuration could be somewhat different). This is less significant over Greenland as the PRISM4 Greenland ice sheet is based upon the results in Koenig et al. (2015) and supported by

simulations in Dolan et al (2015a), which take into account a variety of model differences for simulations over Greenland. The reconstructed Greenland ice sheet is also consistent with modelling results presented in Contoux et al. (2015) and Yan et al. (2014) and a number of different proxy records for the mid-Piacenzian. For example, Bierman et al. (2014) have shown a preservation of a preglacial landscape which suggests potential subaerial exposure in Central Greenland for up to 1 million years prior to major ice build-up at 2.7 Ma. Evidence of vegetation suggesting ice-free conditions can be found in North

Greenland (Funder et al., 2001), at Ile de France (Bennike et al., 2002), on Ellesmere Island and the Canadian Archipelago (De Vernal and Mudie, 1989; Thompson and Fleming, 1996; Ballantyne et al., 2006; Csank et al., 2011), and these offer limited constraints on a mid-Piacenzian Greenland reconstruction. A recent reassessment of pollen derived from ODP Hole 646B off Southwest Greenland (de Vernal and Mudie, 1989) confirms that southern Greenland may have been vegetated (boreal and cool-temperate conditions) during warm parts of the Pliocene (A. de Vernal, personal communication, 2014).

Analysis of sea-ice at Site 910 (~80°N, 6°E) suggest conditions probably similar to the modern sea-ice summer minimum at this time.

A more thorough analysis of model dependency over Antarctica is currently underway, however, de Boer et al (2015) show that the choice of ice sheet model has a smaller effect than the climate forcing on the reconstructed ice sheet. As the PRISM4 Antarctic ice sheet is not updated from PRISM3 it is necessary to assess whether the reconstruction remains consistent with

more recent proxy records and modelling studies.

The reduction in ice at the Wilkes Land margin of East Antarctica is supported by proxy evidence suggesting a dynamic ice margin during the Pliocene (e.g. Young et al., 2014). Yamane et al. (2015) use cosmogenic nuclide exposure ages to show that the different areas of the East Antarctic ice sheet were hundreds of meters thicker during the Pliocene. Ice sheet modelling presented within Yamane et al. (2015) suggested that the continental interior could have been up to 600 m higher than today,

which is consistent with the increased elevation in our PRISM4 reconstruction.

Moreover, independent simulations presented in Pollard et al (2015), which are broadly representative of a warm Pliocene climate, are consistent with the areas of ice sheet retreat in the PRISM4 reconstruction (albeit to a lesser extent). Pollard et al. (2015) detail new mechanisms for ice-cliff failure and melt-driven hydrofracture at the grounding line of the Antarctic ice sheet that lead to significant retreat into the Aurora and Wilkes subglacial basins as well as over areas of West Antarctica

(producing around 17 m of global sea level rise).

### 4.3 Ocean surface characteristics and implications

The most pronounced large-scale feature of Piacenzian climate reflected in the marine (and terrestrial) realm is a reduced pole to equator temperature gradient, enhanced by polar amplification and the *relative* stability of low latitude SST (Salzmann et





al., 2013; Dowsett et al., 2013a; Haywood et al., 2013). Microfossil assemblages from Piacenzian tropical oceans exhibit a high degree of similarity to extant tropical assemblages. Conversely, mid- to high latitude Piacenzian assemblages appear to be displaced toward the poles relative to present day distributions. These assemblages presumably migrated in response to changes in climate (Saupe et al., 2014). These displaced Piacenzian faunas suggest polar amplification of surface temperature,

in agreement with other independent estimates of Piacenzian SST based upon biomarkers, faunal and floral assemblages, and Mg/Ca palaeothermometry (e.g., Lawrence et al., 2009; Bartoli et al., 2006; Cronin, 1988; De Schepper et al., 2009; Barron, 1992; Knies et al., 2014).

Piacenzian marine records from the Arctic are rare, thus our understanding of environmental conditions in this important region is inadequate.  The polar amplification exhibited in our SST reconstruction is supported by analyses from a large number of

mid- to high latitude localities (Dowsett et al., 2012; Figure 8a). Sites 907, 909 and 911 are the highest latitude sites in the North Atlantic used in our reconstruction and have a large influence on the SST anomaly field for the Piacenzian because of the strong warming indicated at those locations based upon Mg/Ca and alkenone methods (Robinson, 2009). Mattingsdal et al. (2014) revised the age model at Site 911 and it appears that the interval studied by Robinson (2009) now falls outside the PRISM interval. Sites 907 and 909 are located at 69°N and 78°N, respectively. They record SST of ~12°C. However, Lawrence

et al. (2013) noted a sharp decline in productivity in the northern Atlantic starting as far back as 3.3 Ma, with earliest signs of decline occurring at 69°N and progressing transiently to lower latitudes. In addition, Robinson (2009) proposed that the discontinuous temperature data could represent brief pulses of warmth advecting into the Arctic in a cooler background climate state, similar to what has been recorded by Bauch (1999) and Bjørklund (2012) in younger age material.  Yet the warm SST data are in agreement with available terrestrial estimates (Ballantyne et al., 2010; Salzmann et al., 2011).

If temperatures in the far northern Atlantic did begin to cool earlier than the Piacenzian, the very warm SSTs in the PRISM data at these sites could be linked to signal carriers brought in by currents or could be the result of reworked older material. There is not sufficient evidence at this point in time of any such reworking so for now PRISM4 data maintains high latitude warm values at Sites 907 and 909, but community scrutiny of these sites will continue and the PRISM SST reconstruction north of the Arctic Circle should be considered tentative until well-dated, higher confidence marine and terrestrial data are

available.

The routinely used SST proxies in tropical regions all have limitations that may obscure low latitude warming above pre-industrial levels (Dowsett and Robinson, 2009; Dowsett et al., 2013). Faunal assemblage techniques are limited by a calibration to the modern ocean which does not exceed ~30°C. Similarly, the $U_{37}^{K'}$ alkenone unsaturation index becomes saturated at 28°C precluding its use in warm pool regions. While the Mg/Ca palaeothermometer is able to estimate temperatures above 30°C,

successful application requires knowledge of the composition of seawater at the time fossil tests composed of $CaCO_3$ were secreted. In addition, syn- and post-depositional processes may alter the geochemical signal and resulting SST estimates. In a recent study, O'brien et al. (2014) applied another biomarker ($TEX_{86}$) and concluded that previous estimates of Pliocene tropical warm pool temperatures based upon Mg/Ca palaeothermometry may have been underestimated. Dowsett (2007b)



suggested the possibility of western equatorial Pacific conditions warmer than present day based upon structural changes in planktonic foraminifer assemblages. Thus, where the low latitude PRISM SST reconstruction is based upon these proxies, it may be obscuring tropical warming.

## 4.4 Land surface characteristics and implications

Parallel to changes in ocean surface temperatures, the zonation of terrestrial biomes during the Piacenzian also indicates a diminished equator to pole thermal gradient enhanced by polar amplification (Salzmann et al., 2013). The most prominent changes in northern hemisphere Piacenzian biome distribution compared to present day include a northward shift of temperate and boreal vegetation zones in response to a warmer and wetter climate as well as expanded tropical savannas and forests at the expense of deserts (Figure 6a). Climate estimates based on palaeobotanical and multiproxy evidence indicate that mean

annual temperatures in the northern high Arctic were, during warm Piacenzian intervals, between 14-20 °C higher than today (Figure 6a; e.g., Ballantyne et al. 2010; Andreev et al., 2012). There is evidence that greater warming occurred in the winter than during summer (e.g., Elias and Matthews, 2002; Wolfe, 1994; Pound et al., 2015).

A globally wetter climate during the Piacenzian supported the formation of megalakes, including Lake Zaire and Lake Chad in Africa and Lake Eyre in Australia (e.g., Drake et al., 2008; Otero et al., 2009).  Model sensitivity experiments show that the

addition of soils to the new BIOME4 reconstruction notably increased surface air temperatures in Australia, southern North Africa and Asia, whereas megalakes generated an increase in precipitation in central Africa and western North America (Pound et al. 2014).

The PRISM4 terrestrial surface reconstruction has a number of uncertainties, particularly in regions of low data coverage such as central North America, South America and North-West Africa, including the Sahara (Figure 6a). Additional uncertainties

are caused by insufficient age control and resolution of individual palaeorecords (Salzmann et al. 2012). The global reconstruction shown in Figure 3 and Figure 6a, like the SST reconstruction, present "snapshots" of terrestrial environments which did not necessarily co-occur in one orbitally defined "interglacial" or "glacial" interval. High-resolution records available for the Piacenzian (e.g. Andreev et al. 2014; Panitz et al. 2015) indicate large shifts of vegetation zones during warm/wet and cold/dry periods within the PRISM interval (Figure 6a). Although these shifts clearly indicate that the Piacenzian

warm climate was not stable and uniform, it should be noted that the Piacenzian vegetation shifts were minor in comparison with Pleistocene interglacial-glacial fluctuations.

## 4.5 Addressing confidence

### 4.5.1 Chronology

The single most important element of any palaeoenvironmental reconstruction is the confidence that can be placed in the

chronologic framework within which it is created. In a general sense, Pliocene geochronology is stable compared to other parts



of deep time, and the combination of radiometric dating, biochronology, palaeomagnetic stratigraphy and LR04 marine oxygen isotope stratigraphy provides the potential to place all events on the same temporal scale with high confidence.

Ideally, all points in a synoptic reconstruction should represent the same instant in time, however no palaeoreconstruction can actually achieve synchrony. The PRISM time-slab covers six "interglacials" in a ~260 Ky interval (Figure 1). The warm peak averaging technique used for SST estimation strives to develop mean interglacial conditions within the time slab, at each site. There is no *a priori* reason to assume the values obtained are synchronous across multiple sites. Analysis of more recent glacial terminations shows diachronous $\delta^{18}O$ responses between ocean basins of up to 4 Ky making correlations independent of climate signals even more important (Skinner and Shackleton, 2005; Lisiecki and Raymo, 2009). In both the marine and terrestrial realms, syn- and post-depositional processes result in time averaged signals where most of the variance is concentrated at low frequencies, further complicating the synoptic nature of reconstructions (Bradley, 1999).

For the purposes of a conceptual understanding of the mid-Piacenzian, however, this equilibrium climate reconstruction has proven most useful. By providing uncertainty ranges and palaeoenvironmental estimates in terms of cold/dry and warm/wet (terrestrial) and maximum, mean and minimum warming (marine), the data sets provide guidelines and realistic limits for a robust Piacenzian global palaeoenvironmental reconstruction. For the purposes of verifying palaeoclimate model simulations, the time slab would ideally need to collapse (temporally) as much as possible toward a true time slice, like the 3.205 Ma datum proposed by Dowsett et al. (2013b) and Haywood et al. (2013). At that time, correlative with MIS KM5c, orbital forcing was close to present day forcing. In addition, simulations suggest minimal surface temperature changes for 20 Ky before and 20 Ky after isotope peak KM5c (Prescott et al., 2014). Even then, due to the time averaging discussed above, the practice of comparing model output and palaeoenvironmental estimates derived from geological archives still has a number of methodological limitations and uncertainties (Haywood et al., 2016).

### 4.5.2 Palaeoenvironmental estimates

The confidence we have in the PRISM4 reconstruction and in each of its components speaks directly to its value in advancing our understanding of past warm climates as well as our ability to verify model simulations. Salzmann et al. (2013) assessed the confidence placed in terrestrial palaeoenvironmental estimates (surface air temperature and precipitation) for terrestrial localities included in PRISM3. Dowsett et al. (2012) attempted to place a semiquantitative estimate of confidence (λ) on the marine localities and materials they yielded for palaeoenvironmental estimation, as well as on the performance of the techniques used to form temperature estimates. These λ values allow others to know how confident, in a relative sense, those most familiar with the data are of the process. While a step in the right direction, λ does not speak to calibration errors nor impacts of processes that cannot be quantified. For example, under a certain level of post-depositional carbonate dissolution, fragile thermophillic taxa may be preferentially removed from a planktic foraminifer assemblage, making the resulting assemblage appear to come from a colder environment. Quantifying the degree of dissolution, or in some cases whether dissolution has taken place, is difficult at best. Also, regardless of fossil group or technique, all methods are in effect analog methods and require an assumption of stationarity. While it is possible to measure the impact that drift in environmental





preferences of a particular taxon has on the resulting environmental estimate, it is not necessarily possible to quantify the degree to which drift took place. Therefore, these confidence schemes are hybrids between those parts of the environmental estimate (e.g., temperature) that can be quantified and those based upon expert judgment.

### 4.5.3 PlioMIP2 data/model comparisons

A first order outcome of the initial PlioMIP data/model comparisons (Dowsett et al., 2012; 2013a; Salzmann et al., 2013) was to show that using palaeoenvironmental estimates that represent a time averaged equilibrium climate to verify simulated conditions representing a specific temporal horizon is problematic. In an effort to reduce uncertainty, PlioMIP2 experimental protocols now call for sub-orbital resolution *verification* data around a specific stratigraphic target of interest that exhibits orbital forcing close to that of present day (Dowsett et al., 2013b; Haywood et al., 2013). An initial time slice centered at 3.205

Ma (Figure 1) has been adopted by the community for the generation of short time series to be used for comparison with PlioMIP2 experiments (Haywood et al., 2012; 2016).

PRISM4 is generating short high-resolution time series between MIS M2 and MIS KM3 (Figure 8). Where independent means of correlation are available through the oxygen isotope record, some sequences will achieve suborbital chronologic resolution. This will allow for a more direct comparison of change at multiple sites from approximately the same time.

Time series with orbital scale chronological resolution sufficient to provide estimates within the time slice window are not abundant on a global scale and therefore data/model comparisons will need to have a regional focus. Alternatively, transient experiments provide a method for comparison of model derived palaeoclimate variables and time-series developed from geological archives. Comparing transient simulations to time series will provide much needed information on the variability and dynamics of Piacenzian climate change and may have less of a requirement for synchrony. Figure 8b shows the location

of some of the initial PRISM4 time series for the North Atlantic region. Two of these sequences also carry high-resolution pollen records which allow marine-terrestrial correlation and comparison with simulations (Panitz et al., 2015).

### 4.5.4 Palaeoenvironmental data integration

Localities suitable for high-resolution marine-terrestrial correlation (Figure 8b) also lend themselves to a different approach to palaeoenvironmental reconstruction, which in turn provides an alternative for data/model comparison. Dowsett et al. (2013b)

outlined challenges to conventional palaeoclimate reconstruction citing as rationale information loss due to underutilization of multivariate data sets capable of a more holistic reconstruction. They called for new methods focused on regional and/or process-oriented reconstructions similar to those being developed for PRISM4. This palaeoenvironmental data integration (PDI) moves away from single variables like temperature and uses the full potential of palaeontological and geochemical data to provide a more nuanced holistic palaeoenvironmental reconstruction.

These regional and process-oriented reconstructions are useful both in terms of our conceptual understanding of Piacenzian conditions and as the basis for a more robust verification of models. Rather than compare a single variable (e.g., temperature) in isolation, Earth System Models can use a variety of output to simulate palaeoenvironments and compare those with



Piacenzian reconstructions that are strengthened by multiple lines of independent geological evidence. Placing error bars on individual palaeoenvironmental estimates (e.g., surface temperature) generally only takes into account a small number of potential contributions to uncertainty. The overall agreement between model and data, using multiple environmental parameters, would be a strong indication of robust model performance. It is otherwise possible to have univariate agreement

for the wrong reason. Stated differently, it is conceivable for both model and data to produce an identical surface temperature, but under completely different environmental conditions. Thus, the PDI approach would provide an alternative indication of the models predictive abilities.

## 5. Summary and conclusions

We have assembled an integrated series of data sets that together reflect our current best understanding of the features of the
Pliocene (mid-Piacenzian) world. The PRISM4 reconstruction indicates reduced pole to equator temperature gradients with polar amplification due in large part to reduced sea-ice extent. The current reconstruction contains, for the first time, an independent elevation (and bathymetry) data set that takes into account GIA and dynamic topography. This reconstruction suggests shoaling in the Indo-Pacific through-flow region, a closed Bering Strait, and a much-reduced connection between the Atlantic and Arctic Oceans. The inclusion of soils and lakes increases the diversity of information provided for the terrestrial
realm. These are both the result of, and forcing agents for, a warmer and wetter Piacenzian.

There is an inverse relationship between the acuity with which we understand the palaeoenvironment and the degree of confidence we have in that understanding. Future PRISM4 work is focusing on high-resolution sequences including but not limited to palaeontological data preserved in marine and terrestrial sediments with attributes that allow correlation to the geomagnetic polarity time scale and/or marine isotopic stages, and unique settings with excellent age control. The new regional
and process-oriented approach will make use of the full potential of palaeontological and geochemical data without over-interpreting temperature or precipitation, to provide a more confident, nuanced holistic palaeoenvironmental reconstruction.

Piacenzian data, including PRISM data, will be used to verify the next set of PlioMIP2 palaeoclimate simulations using several new developments. Conventional data/model comparison will be realized using samples from high resolution time series that intersect the PlioMIP2 time slice (3.205 Ma) and, by comparison of time series to transient model experiments between 3.3
Ma and 3.2 Ma. An ever-increasing number of model simulations will enable us to cross-check our interpretations and develop palaeoenvironment settings. This type of locality and region based reconstruction, while not readily conforming to conventional methods of data/model comparison, does provide constraints on simulated climates both in time and space.

Given the high priority of understanding the mid-Piacenzian within the palaeoclimate community, we anticipate a combination of conventional and holistic approaches will provide enhanced understanding of the time period and a better and independent
assessment of palaeoclimate model performance.





## 6. Data availability

All elements of the PRISM4 reconstruction are available from the data section of the PRISM web page: http://geology.er.usgs.gov/egpsc/prism/4_data.html.

**Acknowledgments**

HD, MR and KF are supported by the US Geological Survey Climate and Land Use Change Research and Development Program. AD, AH and SH acknowledge that this research was completed in receipt of funding from the European Research Council under the European Union's Seventh Framework Programme (FP7/2007-2013)/ERC grant agreement no. 278636. US, AH and MP acknowledge funding received from the Natural Environment Research Council (NERC Grant NE/I016287/1). MC is supported by the NASA Modeling, Analysis, and Prediction program (NASA Grant NNX14AB99A) and the NASA High-End Computing (HEC) Program through the NASA Center for Climate Simulation (NCCS) at Goddard Space Flight Center. We thank Jerry Mitrovica for GIA calculations, and Alessandro Forte and Rob Moucha for the dynamic topography change estimates. We thank Daniel Hill, Stephen Hunter and Linda Sohl for helpful input. HD, AD, AH, US and MP also thank the EPSRC-supported Past Earth Network. This research used samples and/or data provided by the International Ocean Discovery Program (IODP), Ocean Drilling Program (ODP) and Deep Sea Drilling Project (DSDP)





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





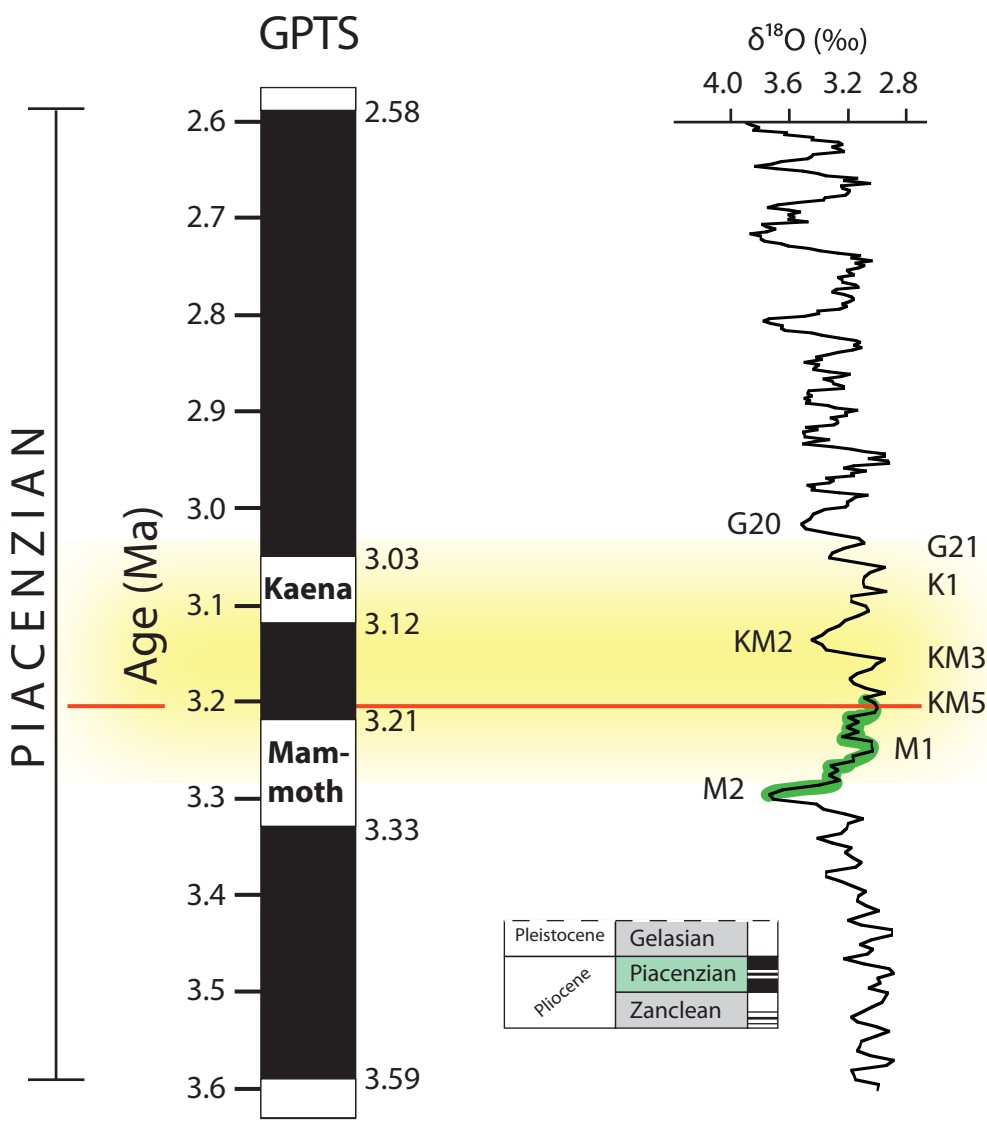

**Figure 1: PRISM Pliocene magnetobiostratigraphic and stable isotopic framework. Benthic δ$^{18}$O record from Lisiecki and Raymo (2005). Position of PRISM3 time slab (yellow band) and extent of PRISM4 time series (green highlighted section of LR04) between MIS M2 and MIS KM5. PlioMIP2 time slice (3.205 Ma) shown by horizontal red line. Magnetic polarity ages from Lourens et al. (1996)**





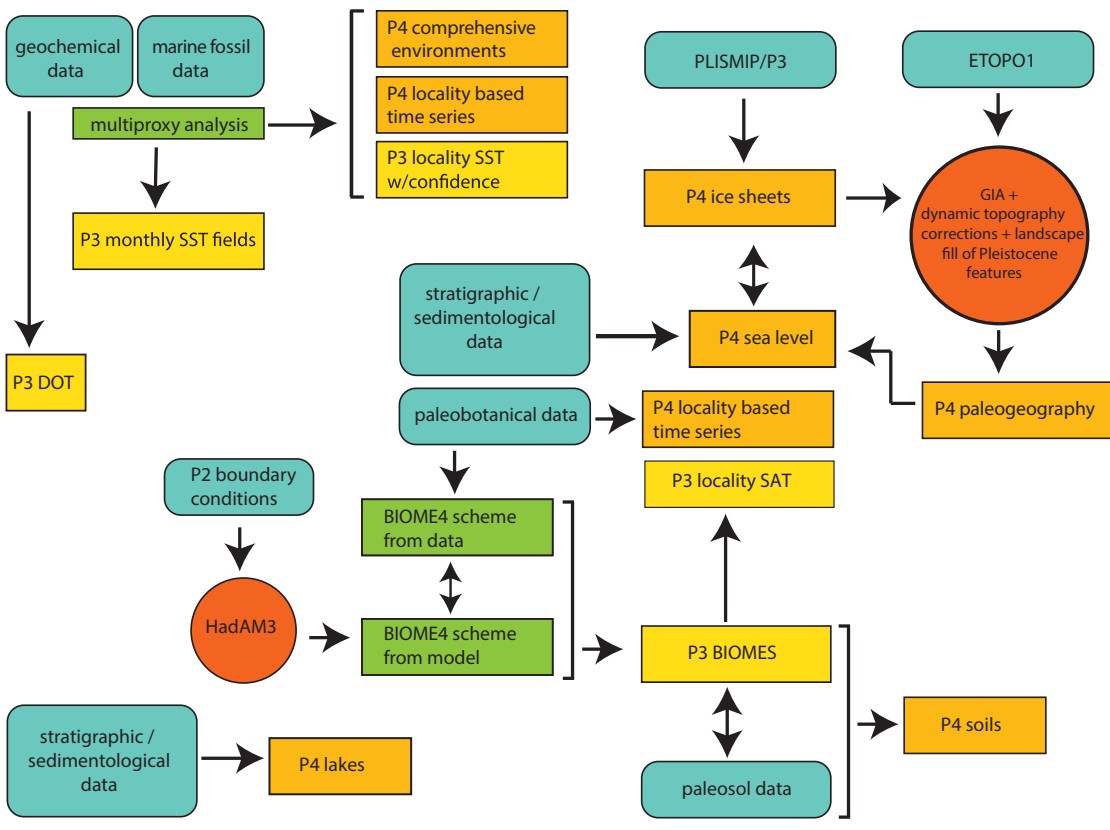

**Figure 2: Interrelationships between PRISM3 and different parts of the PRISM4 reconstruction. Twelve data sets representing components of our Piacenzian reconstruction are shown in yellow and orange rectangles (PRISM3 and PRISM4 respectively). Starting points in the generation of these data sets are indicated by teal rounded rectangles. These are the basic geochemical, faunal, floral, soil, cryospheric, topographic, bathymetric, sedimentologic and stratigraphic data. Marine temperature estimates (SST and DOT) are based upon multiple proxies (including faunal, floral, geochemical and biomarker analyses). Terrestrial vegetation data are translated into a BIOME4 scheme and compared to an independent BIOME4 landscape produced by the HadAM3/TRIFFID model (red circle) initialized with PRISM2 boundary conditions. Integration of data and model output results in the hybrid PRISM3 BIOME reconstruction (Salzmann et al., 2008). Antarctic and Greenland Ice Sheets are based upon PRISM3 and PLISMIP results respectively. Palaeogeography is based upon an initial ETOPO1 digital elevation model incorporating PRISM4 ice sheets, GIA and adjustments due to mantle convection (red circle). Soils are determined through comparison of sedimentological and stratigraphic data with the PRISM3 BIOME reconstruction. Lakes are determined from stratigraphic and sedimentological data (Pound et al., 2014). Sea level is taken from an independent analysis of stratigraphic and sedimentological data (Miller et al., 2012), which is consistent with the PRISM4 ice sheets and palaeogeography presented here.**





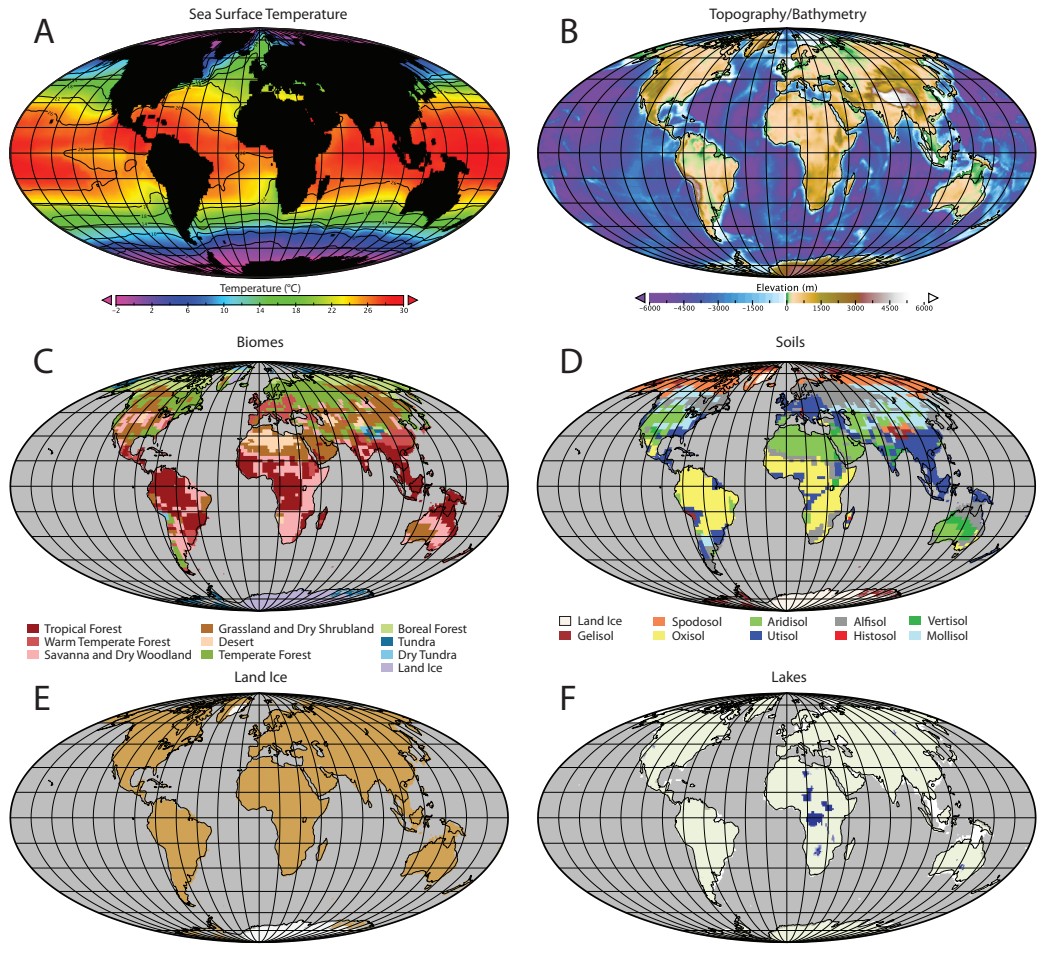

**Figure 3: PRISM4 reconstruction. (A) sea-surface temperature, (B) topography and bathymetry, (C) biomes, (D) soils, (E) land ice, (F) large lakes.**





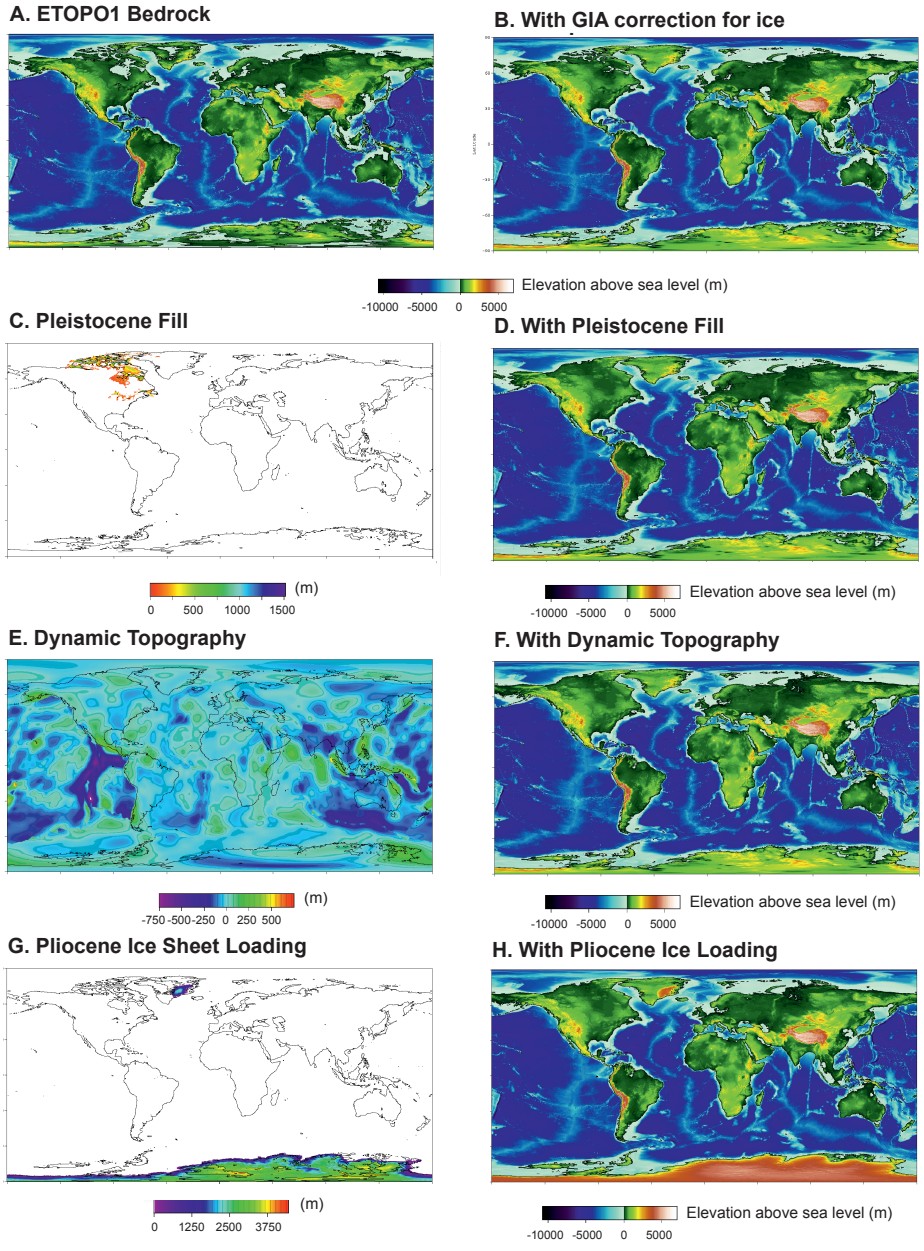

**Figure 4: Generation of the PRISM4 palaeogeography. (A)** shows 'bedrock' (base of the ice sheets) version of ETOPO1 sampled at the working resolution of a quarter degree; **(B)** GIA correction; **(C)** Pleistocene landscape fill; **(D)** Pleistocene landscape features filled in; **(E)** dynamic topography correction; **(F)** Dynamic topography correction applied to (D); **(G)** Piacenzian ice sheets; **(H)** Retrodicted PRISM4 palaeogeography ((G) applied to (F)).





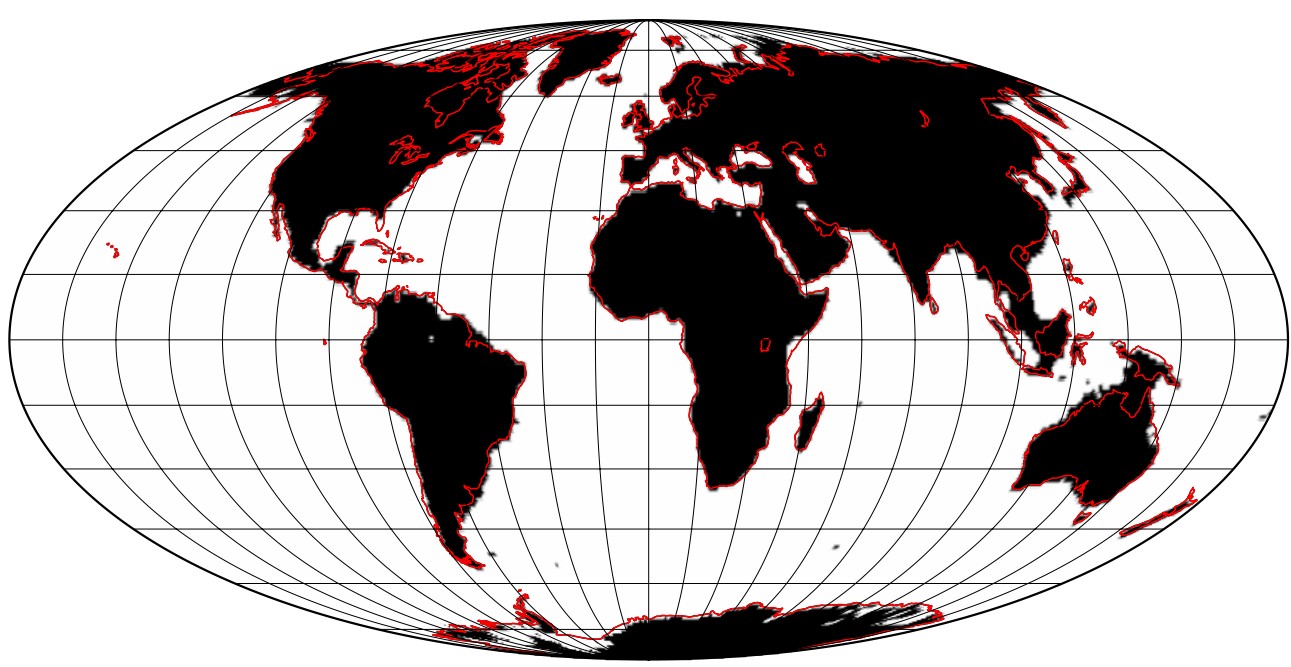

**Figure 5: Pliocene sea level. Distribution of PRISM4 (Piacenzian) land area shown in black. Modern (ETOP01) coastline (red) for comparison.**



A

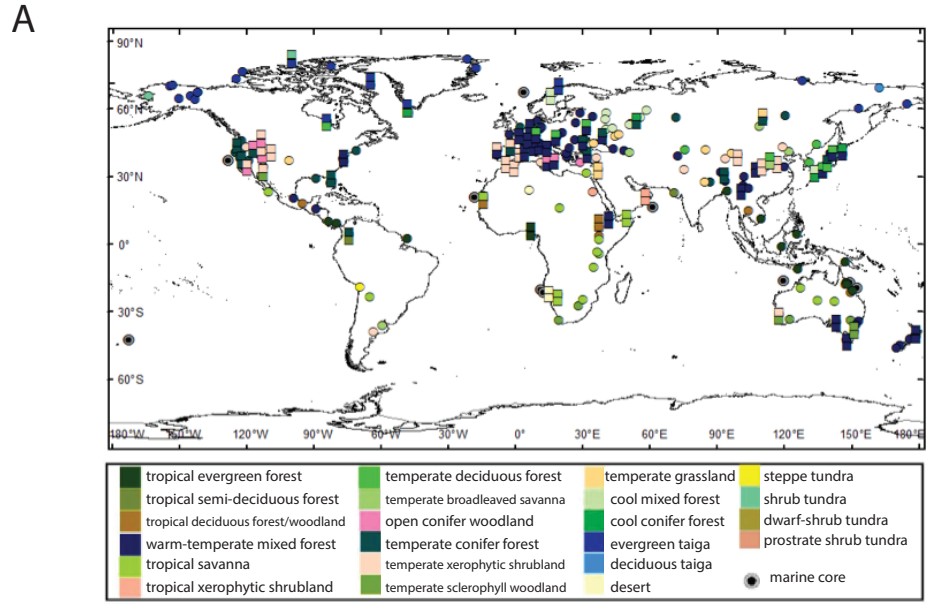

B

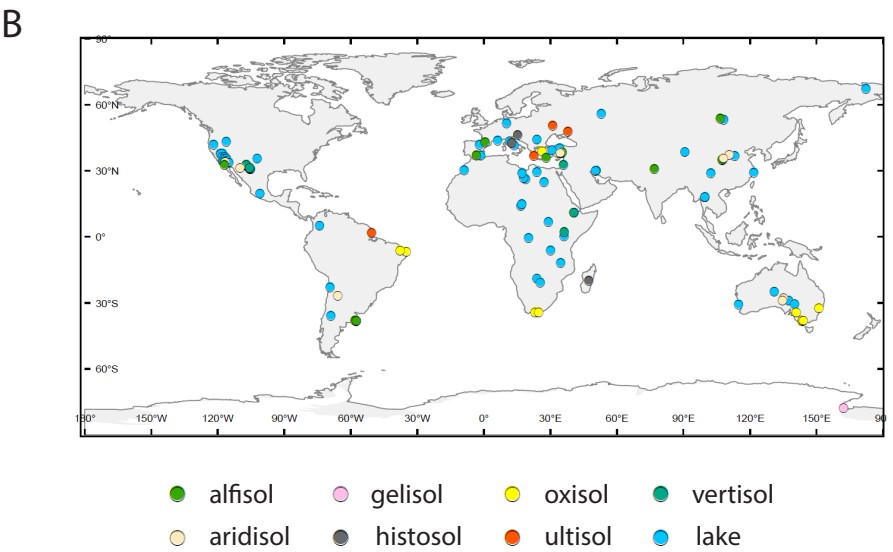



**Figure 6: (A) Vegetation reconstruction for a cold/dry (upper square) and warm/wet (lower square) period of the Piacenzian (3.6-2.6 Ma). Circles show dominating Piacenzian biomes for each palaeobotanical site. For a full explanation of methods and list of literature for each site, see Salzmann et al. (2008, 2013). (B) Geographic distribution of Piacenzian soil and lake localities used in the global reconstructions. Circles show either the dominant soil type reconstructed from palaeosol occurrences or the geographic center of lakes. For a full explanation of the method and list of literature for each site, see Pound *et al.* (2014).**

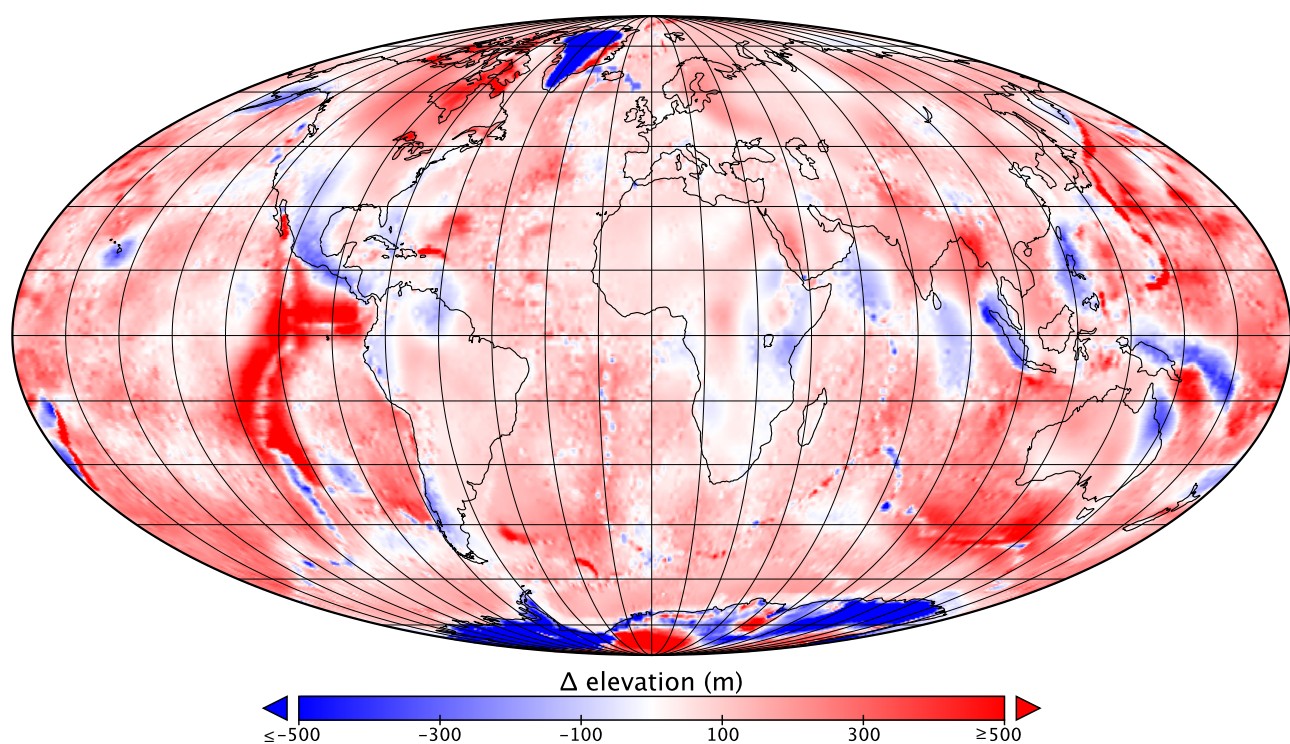

**Figure 7: PRISM4 palaeogeographic anomaly showing Piacenzian elevation minus modern (ETOPO1) elevations (red positive, blue negative).**





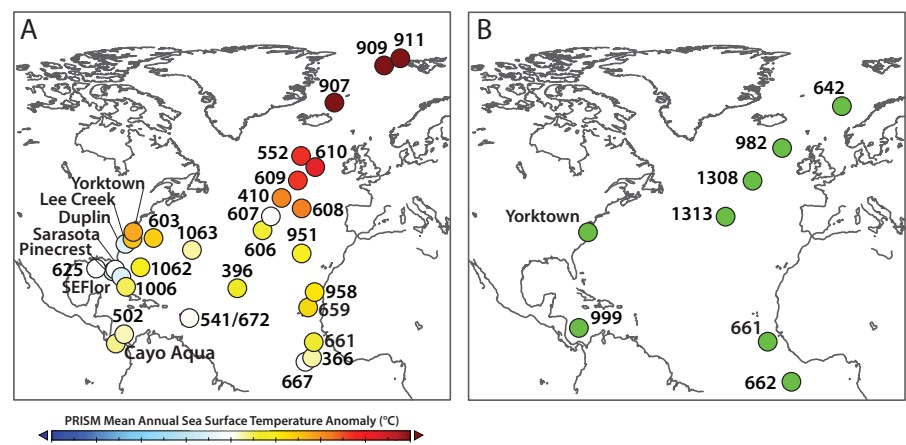

**Figure 8: PRISM North Atlantic region data. (A) Map of PRISM3 North Atlantic region SST anomalies (From Dowsett et al. 2013a). (B) Location of PRISM4 sites which will have integrated paleoenvironmental interpretations and time series.**



**Table 1**. Components of PRISM4

| Component | Reference | Filename |
| --- | --- | --- |
| Palaeogeography | Rowley et al., 2013 | [Plio_enh_topo_v1.0.nc] |
| Land-Sea Distribution | Rowley et al., 2013 | [Plio_enh_LSM_v1.0.nc] |
| Terrestrial Biomes | Salzmann et al., 2008; 2013 | [Plio_enh_mbiome_v1.0.nc] |
| Ice Sheets | Dolan et al., 2015a | [Plio_enh_icemask_v1.0.nc] |
| | Koenig et al., 2015 | |
| | Hill, 2009 | |
| Soils | Pound et al., 2014 | [Plio_enh_soil_v1.0.nc] |
| Lakes | Pound et al., 2014 | [Plio_enh_lake_v1.0.nc] |
| Sea-Surface Temperature | Dowsett et al., 2009; 2013 | [PRISM3_SST_v1.0.nc] |
| Deep Ocean Temperature | Dowsett et al., 2009 | [Global_dot_v2.0.nc] |