# Peer review of "The PRISM4 (mid-Piacenzian) palaeoenvironmental reconstruction"

_Climate of the Past, 2016_

## Referee Comment (RC1) · Anonymous Referee #1 · 23 May 2016

The manuscript by Dowsett and co-authors presents the new PRISM4 reconstruction of environmental conditions during the mid Piacenzian. The data sets have global reach, and combine and update previous syntheses of sea surface temperature and vegetation, alongside land and sea ice extent. For the first time the PRISM synthesis includes soils and lakes, as well as new paleogeography calculations. The paleogeography calculations incorporate a number of important characteristics, including the potential impacts of glacial isostatic adjustment and Earth's dynamic topography on the Earth surface relief. It is important that these variables are taken into account in reconstructions and models of past climates which pre-date the large glacial/interglacial cycles of the Quaternary period, since using modern topography (shaped by those large ice sheets) could influence both regional climates and the inception or expansion of ice sheets. The manuscript is well-written, dealing with each of these new considerations

in turn, and outlining how the assumptions for the modelling components may be supported by existing data where available. The PRISM4 synthesis is a valuable contribution to how we understand past warm climates, and previous versions have proved to be fundamental to robust data-model comparisons. The new information outlined here moves this forward by explaining and incorporating some additional likely influences on late Pliocene climate, which are important if we are to better understand the controls and feedbacks operating at this time.

Although it is definitely valuable to show the PRISM4 reconstruction at a global scale (Figure 3), both the text (e.g. Discussion section 4.1) and the figures themselves indicate that there are some strong regional impacts of the new paleogeography which are not clearly visible on Figure 3. This is particularly for the North Pacific and Antarctic regions, where the projections in Figure 3 make it difficult to see the changes in land-sea distribution, orography, and ice-sheet extent, yet Figures 4, 5 and 7 demonstrate the impact of the new calculations and show that there are strong regional effects. Likewise, the Greenland ice sheet configuration in Figure 3E is difficult to identify. Since the configuration of the Bering Strait and Canadian archipelago are discussed (and are considered to have impact on other climate variables – page 9-10) I recommend that either a polar projection (northern hemisphere) or North Pacific projection is added to the manuscript, containing some of the information of Figure 3 (e.g. SST, biomes, land ice) and perhaps a comparison of the information contained in Figures 4A and 4H or Figure 7 (i.e. the overall impact of the new paleogeography). The authors might also consider whether a similar projection for Antarctica would help their discussion of different evidence for ice elevation change and/or retreat (page 11) since this is very difficult to see in Figure 3. Figure 8 demonstrates that regional-scale information can be very valuable for being able to see the patterns in the data more clearly (although Figure 8 does not seem to be plotted on the new paleogeography?).

Page 11 lines 21-25: I found the insertion of the Yamane et al. (2015) discussion confusing, lying between a statement about reduced ice at the Wilkes Land margin

and a subsequent note (line 26 onwards) about areas of ice sheet retreat. The text may flow better if the evidence for increased ice elevation in some areas is stated before discussing the Wilkes Land retreat?

Page 12: there is a good discussion about the different issues which can be associated with using a variety of biological proxies. Does the PRISM4 reconstruction include all of these different proxies, or have certain proxies been selected? For sites with multiple SST data, how did the authors incorporate the results into the synthesis in Figure 3 where the proxies did not give the same values? Table 1 is cited for the original data sets, but this table redirects you to the synthesis paper from the information on the original data can be determined. It doesn't make it easy to know which proxies have been incorporated into the syntheses.

Figure 2 caption: what is 'DOT' ? I couldn't find the term used elsewhere in the text.

---

## Referee Comment (RC2) · Anonymous Referee #2 · 27 May 2016

This manuscript presented an updated paleienviromental reconstruction of the Pliocene which is called PRISM4 datasets, that will serve as boundary conditions for the numerical models to simulate the Pliocene climate. The information on the reconstruction data source and methods are documented in detail, the differences between the previous data version PRISM3 are well reported and explained. The authors also discussed the major characters and uncertainties during Pliocene such as the closure of seaway, higher topography, smaller greenland ice-sheet coverage etc. The datasets are valuable for the paleoclimate modelling community and they have already been available online.

I have few specific comments below.

1. Based on the discussion of chronology, the authors proposed a 'new' name mid-

Piacenzian to replace the previous mid-Pliocene, because this would appropriately represent the time interval 3.264 Ma and 3.025 Ma as shown in Fig.1. The authors already clearly claimed this reconstruction is being used as boundary condition for PlioMIP. The new name is not convenient for modellers to present their results and interpret model-data comparison. It will cause chaos to mention Pliocene, mid-pliocene and now mid-Piacenzian. Maybe it is more precise on the consideration of geological chronology, but scientifically it does not mean anything new. I would suggest to keep mid-Pliocene and add a precise number like 3.025 Ma, as is already mentioned frequently in the manuscript.

2. The most important information of the reconstructions are illustrated in Fig.3. These are global features, but the map projection makes it difficult to observe the details, especially those over the polar region. I suggest author to use other map projections e.g. Robinson to enable the observation for polar region, the same projection should be applied to Fig.5 and Fig.7. I also suggest that Fig.3 use the PRISM4 coastline as shown in Fig.5, not the modern coastline.

3. Page 8, section 3.4, Ocean temperature and sea ice, here 'ocean temperature' should be sea surface temperature. And sea ice field is missing in Fig.3, it is needed for the atmospheric model as ocean boundary condition. Since sea ice in a warm climate is a major focus for the coupled model simulations, I expect there would be more description and discussion on sea ice even though it remains unchanged from the PRISM3 reconstruction.

4. Page 8, section 3.5, in line 11-12 mentioned 'surface temperature and precipitation anomalies', these are certainly interesting climate parameters and should be included in reconstructions to enable the model-data comparison. If these reconstructions are not good to use, tell us why.

5. Page 8, section 3.6, line 23, "Based upon colour and texture each soil can be assigned an albedo value", when the soil type is used in the model, surface albedo

definitely needs to be assigned, for the PlioMIP, it would be good the assigned albedo is given together with the soil type, to avoid that different model group will assign different values.

6. The detailed descriptions on paleogeography are interesting, as shown in Fig.4 and Fig.7, as mentioned in page 10 line 20-22, there are some interior continental region has decreased the elevation but did not explain why, is it due to the dynamic topography that showed in Fig.4E? A few hundred meters decrease at the costal area would have large impact on low level flow and thus the rainfall pattern, therefore it is important to make sure that these decrease in topography elevation is not artificial.

7. High-resolution time series data in north Atlantic are mentioned in section 4.5.3, and the locations are shown in Fig.8b, it would be interesting to have one figure to show the evolution of these time series and gain some impression on the variability.

8. The provided online SST data does not follow the PRISM4 land-sea mask.

9. Quality of Fig.6 needs to be improved, Fig.4 should be larger.

Typing errors:

1. Page 3, line 3, 'the then inability', remove 'then'

2. page 11, line 2, 'modeled output' change to 'model output'

---

## Author Comment (AC1) · 28 May 2016

We appreciate the comments and suggestions made by this reviewer.

We agree that our choice of map projections make it difficult to see some of the changes we are discussing. We will add a polar view(s) as suggested so the North Pacific, Greenland and Canadian archipelago regions are more easily seen.

Thank you for noticing figure 8 is not plotted using the new palaeogeography. This will be changed and we will make sure all figures are using the new palaeogeography.

We agree the placement of Yamane et al. (2015) should be changed and will rewrite so that the evidence for ice elevation is discussed before the Wilkes Land retreat.

The PRISM4 reconstruction uses the PRISM3 SST reconstruction which is based upon

multiple proxies. The synthesis paper we point to (Dowsett et al. 2012) has this paragraph:

"Different palaeotemperature proxies measure different aspects of temperature by sampling the marine environment at various spatial and temporal resolutions, further complicated by effects unique to each signal carrier and method. Therefore, our multiple proxy analysis is done on a site-by-site basis, taking into account the full range of palaeoenvironmental information derived from a complete assessment of a fossil assemblage and allied geochemical proxies, to determine the overall quality of the temperature estimate. Slight differences between multiple-proxy estimates from a single site strengthen the confidence of the overall site estimate, compared with an estimate from a single proxy."

In addition, Supplementary Table 1 of Dowsett et al. (2012) lists all the original references for the individual temperature estimates. It may be possible for us to include a supplement to this manuscript which repeats this information but also explicitly states which types of data are considered at each locality.

Thank you for catching our use of DOT without first spelling out its meaning: Deep Ocean Temperature. We will change this in our revision.

---

## Author Comment (AC2) · 30 May 2016

We appreciate the comments and suggestions made by this reviewer.

We feel the reviewer's comments suggest the PRISM4 reconstruction only as boundary condition data for paleoclimate models. We explicitly state "These reconstructions serve two purposes: to assemble the best information possible to provide a conceptual model of the Piacenzian palaeoenvironment, and to provide the data as quantitative, gridded arrays to the palaeoclimate modeling community for global climate model simulations." Our responses to the reviewers numbered comments are made with the understanding that PRISM4 is not simply a data set for PlioMIP2.

1. We respectfully disagree with this comment. The PRISM interval has always been within the Piacenzian Stage. However, our decision is necessary in order to correctly

address the new "official" stratigraphy of the Pliocene (enacted by the International Commision on Stratigraphy), which moved the Piacenzian from the mid to the late Pliocene. This decision made the former division into an early (Zanclean), mid- (Piacenzian) and late (Gelasian) Pliocene obsolete. Calling the Piacenzian still "mid-Pliocene" is not only even more confusing, it is also scientifically incorrect.

2. We thank the reviewer for this comment on map projections. We agree that our choice of map projections obfuscates the information we are trying to display. We will add map views and projections that make it easier to discern changes made to the North Pacific, Greenland and the Canadian Arctic.

3. As the reviewer suggests, a more appropriate heading for section 3.4 is "Sea Surface Temperature and Sea-Ice." We have changed this in the manuscript and will add information on sea-ice. We will explicitly state, as we did for previous PRISM reconstructions, that PRISM displays sea ice as SST = -1.8°C. This is easier to see in the PRISM3 monthly SST reconstruction. We can provide the sea ice limits as a dashed contour and place these PRISM3 monthly SST maps in the supplement.

4. We agree with reviewer#2 that the terrestrial temperature and precipitation estimates and anomalies are "certainly interesting parameters". The PRISM4 Biome reconstruction (Figures 3 and 6a) is based on 208 palaeobotanical sites and has been updated from Salzmann et al. (2008, 2013) and Dowsett et al.(2010). Surface temperature and precipitation anomalies have been derived from literature, from multi-proxy temperature reconstructions or by applying the Coexistence Approach (Mosbrugger and Utescher,1997). All numerical climate estimates including a discussion on relative confidence for each locality and uncertainties can be accessed in Salzmann et al (2008 and 2013). As previously stated this manuscript focusses on the PRISM4 reconstruction and not on climate modelling or data-model comparison. We therefore prefer to keep our paper focused and not include already published data and discussion. However, in order to address reviewer#2' concerns, we reworded this section to make clearer where climate estimates and anomalies can be accessed.

5. We originally included the albedo values for the different soil types used in PRISM4 but decided to remove them since they were provided in Pound et al. (2014). We agree that different modeling groups might assign different values, but this is an issue for PlioMIP2 and not the PRISM4 reconstruction. We note the albedo values are also provided in the PlioMIP2 experimental design paper included in this theme issue, but we will add the albedos to the supplement for completeness.

6. We are not sure how to respond to the reviewer comment about "being sure these changes are not artificial." The steps used to create our palaeogeography are clearly stated and shown in the figures. The method is reproducible. Whether every change, in any PRISM data set, is representative of the actual mid-Piacenzian state, is impossible to ascertain. PRISM is a conceptual model of mid-Piacenzian conditions. It is undoubtedly incorrect in many places and over the years has been modified when new and better data became available. At present, we feel the new PRISM4 palaeogeography is a major improvement over that used in previous PRISM reconstructions. It is consistent with the limited data available for palaeotopography, and is a useful working hypothesis for mid-Piacenzian conditions.

7. We thank the reviewer for this comment and agree that it would be beneficial to include a figure showing at least one site with high resolution data. We will include a figure and some discussion of the variability in our revision.

8. The PRISM4 reconstruction uses the PRISM3 SST fields, unchanged. As a conceptual model the difference in coastline between PRISM3 and PRISM4 is negligible. We can extend the mean annual SST field in the figure so that it matches the PRISM4 coastline, but we cannot make changes to the existing PRISM3 data. If the reviewer is thinking in terms of PlioMIP2, those experiments do not use SST as a boundary condition and, it is never appropriate to use the highly interpolated and extrapolated SST fields for data model comparison (though that has unfortunately been done by those unfamiliar with the data).

9. We appreciate the comment on the quality of Figure 6 and the size of Figure 4. All figures will be revised based upon both reviewers comments.

The typing errors noted on pages 3 and 11 will be changed as suggested.

---

## Author Response (AR1)

Prof. Harry J. Dowsett
Eastern Geology and Paleoclimate
 Science Center
US Geological Survey, MS 926A
12201 Sunrise Valley Drive
Reston, VA 20192  USA

Wing-Le Chan,
Editor, *Climate of the Past*

Wing,

I apologize for the delay in returning this manuscript to you. While the changes were minor and accomplished very quickly, getting in touch with all co-authors for approval of changes was difficult.

We have resubmitted our manuscript after making minor revisions suggested by the reviewers. I summarize those changes below. The USGS has it's own review process and the manuscript was also seen and reviewed by Thomas Cronin and Michael Toomey of the USGS Reston Office. The USGS has approved the manuscript.

I need to point out that we added three authors: Jerry Mitrovica, Robert Moucha and Alessandro Forte.  I did not realize how large a role they played in the development of our new palaeogeography directed by David Rowley. All three have been involved in the revision and I believe their input greatly enhances the PRISM4 reconstruction.

Our sea level equivalent is based upon the ice volume estimates we use. These are in agreement with our new palaeogeography. Some of us felt our sea level equivalent being very close to the Miller et al. (2012) estimate was worth pointing out. Others of us do not believe the Miller et al. estimate is comparable since they did not use a dynamic topography correction. We revised the sea level section and removed the inference that our sea level estimate is comparable to Millet et al. (2012).

All four reviewers indicated a different map projection and possibly a polar view would be helpful. We switched to a Robinson projection wherever possible and also added north and south polar views to the maps in Figure 3.

We accepted all the minor text corrections suggested by the reviewers.

Reviewer 1 noted we did not use the new palaeogeography in figure 8; This has been corrected and we now use the new palaeogeography for all figures.

In response to the question about sea ice we have added the PRISM3 monthly sea ice

cover maps to the supplement.

Reviewer 2 suggested a different heading for section 3.4 and we adopted the new heading.

We added a table of soil albedo values to the supplement.

Reviewer 2 and one USGS reviewer asked about the high resolution PRISM4 time series work. As an example of the types of data we are generating we have included in the Supplement faunal census data and a number of palaeoenvironmental proxies developed for ODP Site 982.

We have increased the quality of Figure 6 and all the figures wherever possible.

We hope you find our revision acceptable for publication.

Best,
* * *
Prof. Harry J. Dowsett

**Lead Scientist, USGS PRISM Project**
Co-Leader, PlioMIP
Journal Editor, *Micropaleontology*
Visiting Professor of Paleoceanography, University of Leeds
Adjunct Faculty, AOES, George Mason University

hdowsett@usgs.gov
571-294-4318